# Evaluating the Implementation of a Pilot Quality Improvement Program to Support Appropriate Antimicrobial Prescribing in General Practice

**DOI:** 10.3390/antibiotics10070867

**Published:** 2021-07-16

**Authors:** Ruby Biezen, Kirsty Buising, Tim Monaghan, Rachael Ball, Karin Thursky, Ron Cheah, Malcolm Clark, Jo-Anne Manski-Nankervis

**Affiliations:** 1Department of General Practice, The University of Melbourne, Melbourne 3004, Australia; tim.monaghan@unimelb.edu.au (T.M.); Malcolm.Clark@ipn.com.au (M.C.); jomn@unimelb.edu.au (J.-A.M.-N.); 2National Centre for Antimicrobial Stewardship, Department of Infectious Diseases, The University of Melbourne, Melbourne 3004, Australia; Kirsty.Buising@mh.org.au (K.B.); Karin.Thursky@mh.org.au (K.T.); Ron.Cheah@mh.org.au (R.C.); 3The Guidance Group, Royal Melbourne Hospital, Melbourne 3050, Australia; 4North Western Melbourne Primary Health Network, Melbourne 3052, Australia; rachael.ball@nwmphn.org.au

**Keywords:** antibiotics, antimicrobial stewardship, audit and feedback, general practice, general practitioner, inappropriate prescribing, practice manager, prescribing, primary care, quality improvement

## Abstract

Inappropriate antimicrobial prescribing contributes to increasing antimicrobial resistance. An antimicrobial stewardship (AMS) program in the form of quality improvement activities that included audit and feedback, clinical decision support and education was developed to help optimise prescribing in general practice. The aim of this study was to evaluate the implementation of this program (Guidance GP) in three general practices in Melbourne, Australia, between November 2019 and August 2020. Thirty-one general practitioners (GPs) participated in the program, with 11 GPs and three practice managers participating in follow-up focus groups and interviews to explore the acceptability and feasibility of the program. Our findings showed that the quality improvement activities were acceptable to GPs, if they accurately fit GPs’ decision-making process and workflow. It was also important that they provided clinically meaningful information in the form of audit and feedback to GPs. The time needed to coordinate the program, and costs to implement the program were some of the potential barriers identified. Facilitators of success were a “whole of practice” approach with enthusiastic GPs and practice staff, and an identified practice champion. The findings of this research will inform implementation strategies for both the Guidance GP program and AMS programs more broadly in Australian general practice, which will be critical for general practice participation and engagement.

## 1. Introduction

There is global concern about inappropriate prescribing of antimicrobials as this can drive antimicrobial resistance and adverse events including allergic reactions, side-effects and altered microbiome without clinical benefit [1,2,3,4]. This has been highlighted as an issue in Australia, which is in the top 25% of countries with the highest rates of antibiotic prescribing in the community compared with other European countries and Canada [5]. Yet, despite these high volumes of antimicrobial prescribing in the community [5], there are no current models for an effective antimicrobial stewardship (AMS) program in Australian general practice to help optimise appropriate prescribing. 

Interventions such as audit and feedback, clinical decision support, and education, particularly where these interventions are electronically delivered, integrated into practice workflow, provide automated clinical decision support, and stimulate patient discussion have been showed to have effects in reducing antibiotic prescribing in general practice [6,7,8,9]. While reducing overall antibiotic prescribing is important [10], the goal of AMS should be centered around reducing inappropriate antibiotic prescribing. 

Our previous work has explored general practitioners’ (GPs) use of electronic medical records (EMRs) and embedded electronic guidelines to inform decisions around antimicrobial prescribing [11]. Our simulation study of a clinical decision support (CDS) tool which streamlined access to antibiotic guidelines and provided customised information for GPs and patients was acceptable and was likely to fit within the clinical workflow [12]. We have also demonstrated the acceptability of a pilot audit and feedback program called the General Practice National Antimicrobial Prescribing Survey (GP NAPS) which provided feedback to GPs around appropriateness of antimicrobial prescriptions and compliance with guidelines [13]. We identified that improved documentation of the reason for prescription in the EMR and point-of-care access to guidelines [14] would facilitate the use of clinical decision support tools and enable automation of a clinically meaningful passive audit process.

This program of work resulted in the development of Guidance GP, a pilot quality improvement (QI) program consisting of a clinical decision support tool to facilitate access to Therapeutic Guidelines (eTG) [14], which are the nationally endorsed prescribing guidelines, and an embedded audit and feedback tool utilising data extracted from general practice electronic medical records (GP NAPS), to permit customised feedback and education webinars. The aim of this project was to evaluate the implementation of Guidance GP in general practice in order to optimise the program to ensure it meets the needs of GPs and inform strategies for broader implementation.

## 2. Results

### 2.1. Participants

A total of 31 GPs from three metropolitan Melbourne practices (practice size range from five to 25 GPs) participated in the pilot Guidance GP program from November 2019 to August 2020. Of the GPs who participated in the program, 11 participated in three focus groups at the end of the study period (participant number ranged from 2 to 6 in each focus group). The mean GP experience was 17.3 years, with a median of 18 years. Three practice managers, one from each practice, participated in a semi-structured interview. Demographic details of participants are summarised in Table 1.

### 2.2. Context

At the baseline GP NAPS audit, there were 231 antibiotic prescriptions recorded in the two-week period between 14 and 17 November 2019. At the follow-up GP NAPS audit from 17 to 31 August 2020, there were 73 total antibiotics prescribed across the three participating practices. Skin and soft tissue infection was the most common indication pre- and post-intervention; amoxicillin and amoxicillin-clavulanic acid were the top prescribed antibiotics pre- and post-intervention, respectively (see Table 2).

The CDS tool was scheduled to be installed across the three participating practices in early February 2020. However, initial technical issues included testing and implementation of the CDS tool in each practice, followed by the COVID-19 pandemic, restricted the implementation of this part of the program into practices. These barriers were discussed in the GP focus groups and PM interviews.

### 2.3. Focus Group and Interview Findings

Themes were grouped within the Clinical Performance Feedback Intervention Theory (CP-FIT) framework into feedback variables, recipient variables and context variables (Figure 1). We elaborate on these variables below.

#### 2.3.1. Feedback Variables

##### Goal

GPs viewed AMS as important, and receiving audit results and feedback assisted their goal setting and reflection on their practice was considered acceptable.

*“…you realise, ‘Well there’s some things that I’m not going to be able to change, but within these contexts… these are the things that I’m going to target in terms of my own reflective practice’….”* GP4

However, GPs would prefer not to be told what to prescribe, emphasizing that GP performance should not only be about following guidelines, but should also consider patient views and beliefs. There were also concerns that data extracted from the EMR did not reflect the nuances of the clinical decision-making process.

*“…I’d have to admit, I think most of us here would probably prescribe by guidelines, so I think a lot of that non-compliance is attributed to personal interactions with patients and their preference.”* GP1

While QI programs for AMS were acceptable to most GPs, there were tensions about the goal of this study—a program to assist GPs to optimise prescribing as opposed to a tool that could be used “against practices” or to show that general practice was “not performing”. Some GPs were concerned that data could be used as key performance indicators (KPI) linked to funding.

*“…my biggest concern about this is the data and what’s going to happen and be used with our data, because I have a very strong sense that the final destination for this stuff is going to be a KPI for funding practices… for an accreditation or to get your PIP (practice incentive payment)… you have to demonstrate that you’ve got 85% compliance with your antibiotic stewardship...”* GP4

Regardless, AMS programs need to be kept front and center to modify GPs’ antibiotic prescribing behaviour.

*“…I think when the goal of the study is essentially behaviour change for doctors, it’s got to be about keeping antibiotic stewardship sort of front of mind. It’s about keeping it salient. So, any way that I can be reminded throughout the year, if it’s a group-based thing or if it’s a pop-up on Best Practice or if it’s part of the induction when the new doctor starts, whatever it might be, it’s just about sort of constant reminders as opposed to “yes, we’re done now and you don’t have to think about it again”. It’s just keeping in your head.”* GP10

##### Data Collection and Analysis

Some GPs felt that the audit duration with the pop-up reminder function (i.e., where a pop-up box would appear if the ‘reason for prescription’ field was left blank allow a selection of indications from an indication list) should be kept short.

*“I certainly don’t want that big popup window any longer than two weeks…”* GP3

*“I don’t think I needed to be whacked on the nose with the newspaper multiple times...”* GP7

Interestingly, GPs commented that they did not notice the GP NAPS pop up tool as much during the follow-up audit period. This might be due to the change in prescribing habit, or from the impact of COVID-19 reducing respiratory tract infection consultations in practices. Often patients with respiratory symptoms were automatically referred to respiratory clinics at the height of COVID-19 cases, further reducing the numbers of antibiotics prescribed for patients coming in for these conditions.

*“I don’t know if I don’t put a reason in because I wasn’t seeing the patients or because I’m much better at putting a reason in… it might have just been because we weren’t prescribing nearly as many antibiotics.”* GP9

The impact of COVID-19, therefore, has led to concerns that audit results did not reflect their usual practice as disease presentations changed due to the pandemic.

*“…there was actually a whole type of pathology we weren’t seeing at all.”* GP5

##### Feedback (Display and Delivery)

The feedback report provided a comparison between practices, which reinforced to some GPs that while there were room for improvement, prescribing was similar across the practices.

*“It [feedback report] is actually remarkable how similar all three practices actually end up. They’re pretty much within that sort of two thirds appropriate and one third less appropriate… we keep hearing over and over and over again how terrible GPs are and our prescribing of antibiotics is terrible… Sure, there’s maybe some room for improvement and that might come back to some of the data extraction… So, it’s interesting that we’re probably not as bad as we’re painted to be.”* GP9

GPs thought benchmarking was helpful so that practices could compare their performance with other practices. However, benchmarking should be comparing practices with similar characteristics including patient demographics, practice demographics (rural or metropolitan practice), and practice size and billing type. There was also interest in feedback comparing individual GPs. While this was discussed in the development phase of the QI program, individual-level feedback was not possible with the small number of participants in this pilot study.

#### 2.3.2. Recipient Variables

##### Health Professional Characteristics

Knowledge and skills of health professionals in AMS and their attitude to receiving feedback were considered important factors that would influence the perceived benefit of the program and acceptability of the intervention. In terms of knowledge and skills, GPs thought interventions would be beneficial for younger, less experienced GPs, but also for older GPs who might not be as up to date with the current guidelines.

*“I think it’s an interesting topic and I think it’s a really common presentation… with changing guidelines and getting stuck in habits that may have been generated 15… or 18 years ago or whatever, there probably is a lot of benefit for younger GPs who are starting off their prescribing habits, but I think equally there’s probably just as much benefit for older GPs or GPs more experienced who perhaps aren’t as up to date as they could be …”* GP9

GPs had different attitudes towards feedback. One GP had concerns regarding the way data were presented, feeling that the data on appropriateness were presented in a negative way. Others found the feedback helpful and inspired them to reflect on their own practice.

***“****I kind of feel that scooping up the sub-optimal to enhance the inappropriate data, this happens all the time with GP studies. We saw it with the paediatric parent study of parent satisfaction with GPs.”* GP4

*“Yeah, I think it’s useful to reflect and look at ways that we can improve our use of antimicrobials, yeah for sure.”* GP2

#### 2.3.3. Context Variables

##### Organisation or Team Characteristics

Practice culture, where practices are research focused, was perceived to be strongly linked to whether a QI program would be successful or not.

*“As a group they’re highly active and highly engaged in these sorts of programs or projects… Anything they can learn from, they’re very hungry for that and passionate about making sure things are done right.”* PM2

It was interesting to note that GPs’ engagement was seen as the motivation for practice managers to be involved in QI programs. While actively engaged GPs helped with practice managers’ involvement, GPs who were less engaged often left practice managers at a loss as to how to drive the program forward.

*“Going back to it all, I think one of the downsides to it I’ve found is that the doctors just sort of did what they did... I didn’t find that there was a lot of feedback from them in terms of letting me know what was going on…So I think there was a lack of feedback from them to me.”* PM3

From the GP focus groups and practice manager interviews, it emerged that practice managers’ involvement and leadership was critical to the success (or failure) of any program implementation.

*“Absolutely. Taking that leadership, so not just sort of saying, “Hey guys, do you want to do this quality improvement project? Here you are, this is everything, this is what you need to do.” Actually be involved and be able to actually go to them every week and say, “Hey, how are you going with that? What’s been going on? Are you having issues? What’s something that happened good today?” … to be much more of a leader in the project …”* PM3

##### Implementation Process

There were challenges to the program implementation, including accessing computer systems at the practice sites (in person or remotely); coordinating IT support between practice and research team; practice managers’ time needed to coordinate installation process; and testing of the tools before going live. Technical issues identified during the testing phase of the CDS tool, along with the added complications of the study team not being able to engage with practices during the COVID-19 period, led to the CDS component of the QI program not being implemented into the practices.

Computer access was a major barrier to the implementation of the QI program; practice managers needed to find time to minimise GPs’ workflow interruptions.

*“They did start to get a little bit tired of being booted off their computers… we tried to do it at lunchtimes and they would just run late… as GPs do.”* PM1

In addition, practice managers were not comfortable having external parties accessing their computers, and with the need to use remote access due to the pandemic, this added to the burden of the practice IT support and practice managers overseeing the installation process.

*“And obviously for me it was about protecting patient privacy as a practice manager. So just wanting to make sure that everything was going to be above board, which it was. We’ve had a lot of help from our IT people, especially with COVID and all that…”* PM1

*“I think the implementation of IT, that was the biggest thing, but I wouldn’t let anyone else do it. I don’t give other people access to our IT easily, so that was certainly the most time-consuming thing… I wouldn’t feel comfortable with someone else wandering around different machines and turning them on and doing what they wanted…”* PM2

Cost was also mentioned as a barrier to the installation process, time taken for practice managers to coordinate the implementation, practice IT team to assist with implementing the tools, and GPs to test the system.

*“I think the main issue, in regards to setting it up, is that it actually costs our practice manager lots of time, and with all those IT issues, and I think this is something that’s making it difficult for general practices to participate in the high-quality research.”* GP2

## 3. Discussion

This study highlighted many important factors when implementing a QI program in general practice. Data that accurately reflect GPs’ decision-making process; tools that assisted GPs and their practice in providing better health outcome for their patients; and audits and feedback that were robust and provide meaningful information to the GPs were seen as critical to a QI program. Undoubtedly, a positive “whole of practice” approach with good communication between GPs and practice managers; ensuring strong participant engagement throughout the program; having an engaging practice champion who would advocate for GPs and the program; reducing workflow interruptions; and minimise costs for practices could ensure the success of implementing a QI program within general practice.

While the GP NAPS audit data showed an overall decrease in the number of antibiotic prescribed, a slight overall percentage increase in the number of indications documented in the EMR, and an overall percentage decrease in the indication entered in the audit pop up tool, we cannot determine the significance of these results due to the small sample size. However, GPs were responsive to the feedback report provided at the end of each GP NAPS audit period. Similarly, Meeker et al. (2014) found a significant reduction in inappropriate antibiotic prescribing for acute respiratory tract infections when using accountable justification and peer comparison as behavioural interventions [15]. Nevertheless, some GPs were sceptical as to the value of QI programs, apprehensive about the potential for funding structures to move to “pay for performance” and that ‘inaccurate data assessment’ and analysis that did not take into account clinical nuance required for specific patients and factors outside their control could make general practice performance appear sub-optimal. This is of particular concern as perception of inaccurate data collection could lead to barriers in implementing future QI programs. In addition, general practice funding is linked to practice incentive payments which include QI activities and programs, in a time where general practices are under finance pressure, it is therefore an important area to address.

Time constraints for both GPs and practice staff, cost of activities, issues with information management and technology, and a lack of financial incentives were considered as barriers to QI programs in primary care [16,17,18,19]. While we anticipated that practice managers would need to assist with the coordination of installing the GP NAPS and CDS tools, limited practice availability and IT issues hindered the smooth implementation of the CDS tool. Larger practices with GPs sharing rooms became a logistic issue for practice managers to juggle GPs computer access and installing tools on their computers. Remote computer access was limited due to privacy concerns, leaving little room to manoeuvre.

The COVID-19 pandemic during our study period posed further challenges. Melbourne experienced the first lockdown period from March to May 2020, and the second lockdown in August to November 2020. Most face to face visits at general practices became telehealth visits, and patients with respiratory symptoms were diverted to respiratory clinics. During this period, participating practices commented they did not see patients with respiratory symptoms, thereby further reducing the number of consultations and potentially reducing the number of antibiotics prescribed, leading to possible inaccurate data interpretation.

The small number of participating practices was a limitation to our study. Data collected were limited to three general practices in metropolitan Melbourne, therefore our findings may not be generalizable to more diverse settings, including practices in rural or remote areas. In addition, our study may include selection bias as participants were more likely to have an interest in antimicrobial stewardship and QI programs. This study was largely conducted during the COVID-19 pandemic, which was characterised by reduced respiratory tract infections presentations to general practice, significant changes to delivery of care more broadly (with transition to telehealth) and impacted the delivery and likely participation in the quality improvement activity. Despite this, all components of the intervention were delivered. We did not have access to additional funding in order to carry out additional audits after cessation of lockdown in Melbourne. While our study highlighted several challenges both researchers and practices faced when implementing a QI program, some of the enablers to the success of the implementation process was the enthusiasm of participating GPs in the QI program, and the positive attitude of the practice managers, who were the driving force for this study. They constantly liaised with the research team and their GPs enforcing the importance of having a practice champion who can coordinate, communicate effectively with GPs and advocate for the practice and the program. Other studies have also emphasized the importance of a practice champion, whether this be a GP or a practice staff such as a practice manager, as part of a key ingredient to successful implementation of programs [7,18,19,20]. As practice managers are front and center of the practice, and provide important insights into the daily running of the general practice, it is recommended that they are involved in the planning and the development of future QI programs to reduce challenges identified in this study.

## 4. Materials and Methods

### 4.1. The Intervention

The Guidance GP program consisted of four key components, conducted between November 2019 and August 2020:Baseline GP NAPS audit (November 2019) using passively extracted prescribing data; with a customised feedback report supported by a webinar to discuss findings with participating GPs (December 2019).Royal Australian College of General Practitioners (RACGP) accredited Quality Improvement educational activity supported by North Western Melbourne Primary Health Network (NWMPHN; November 2019–August 2020).Implementation of an electronic CDS tool to provide access to the eTG at the point of care (February–May 2020).Follow-up GP NAPS audit (August 2020) with a customised feedback report supported by a webinar to discuss findings.

During the two-week GP NAPS audit period, a data extraction tool was used to collect data from the EMR for each antimicrobial prescription generated. The tool was embedded into the practice EMR and scanned for prescriptions of any antimicrobials. For each prescription, data fields were extracted from the EMR, these included patient height, weight, renal function (eGFR), medical history, reason for prescription, prescription information, pathology and imaging requests and results. If the ‘reason for prescription’ was not entered, a pop-up box would prompt GPs to select an indication from a live and curated indication list [21]. This is a SNOMED mapped curated list of infection and antimicrobial indications utilised in the National Antimicrobial Prescribing Survey program, which is a national program supporting quality audits of antimicrobials in hospitals and aged care homes [5]. The information gathered was used to compare the actual prescription (antimicrobial choice, dose, duration, indication), with the recommended prescription in national guidelines based on the documented reason for prescription, with assessments adjusted to accommodate any patient-specific factors (such as age, renal function, allergy status) and any known pathogens Thus, a prescription may not align with guidelines, but could be judged appropriate based on patient-specific factors. Reporting could therefore include compliance with guidelines as well as appropriateness [22].

The results from the two-week audit period were provided to GPs in a form of a de-identified practice-specific feedback report in order to allow GPs to reflect on their prescribing practice collectively. In addition to the report, a feedback and education webinar was conducted after each audit for GPs to ask clinical questions and learn from others about potential strategies for QI around AMS.

The CDS tool integrated eTG into the practice EMR. The CDS tool streamlined access to the eTG through an icon located on the main screen and prescribing screens of the EMR [12].

GPs who completed the pilot program were provided with 40 Quality Improvement Continuing Professional Development activity points through RACGP. The program included reflection on practice and documentation of strategies that they would introduce to their practice as a result of participating in the program.

### 4.2. Participant Recruitment

Three metropolitan Melbourne general practices were recruited to participate in the pilot study. An expression of interest was sent to practices via the NWMPHN newsletter. The only inclusion criteria were that practices had to be using the Best Practice™ electronic medical record system. Interested practices contacted the researcher (RBn) and a practice visit was organised where the program was discussed with the practice manager and GPs interested in this study. Incentives for participation included a practice incentive payment (AUD $2000), provision of a 12 month subscription to eTG (valued at AUD $378 per GP), and a AUD $100 gift voucher for practice managers for participating in an interview.

### 4.3. Data Collection and Analysis

At the end of the pilot, three focus groups (one in each participating practice) with GPs and three practice manager interviews were conducted to evaluate the acceptability and usability of GP NAPS and the CDS tool, and any implementation barriers and facilitators of the program were discussed via video conferencing in September 2020. RBn conducted all focus groups and interviews. Data from the focus groups and interviews were digitally recorded, and transcribed verbatim. Transcripts were reviewed to remove any identifying information and all GPs, practice managers, and practice names were replaced by pseudonyms or codes. All transcripts were read at least twice and compared to the recording for accuracy.

The CP-FIT was used to provide a framework to analyse the focus group and interview results [23]. This theory was used as our intervention centered around audit and feedback to inform QI and changes to practice, when indicated. CP-FIT builds on pre-existing theories and proposes that effective feedback works in a cycle of sequential processes, and feedback success is influenced and determined by variables such as feedback variables (goal, data collection and analysis method, feedback display and delivery), recipient variables (health professional characteristics and behaviour response), and context variables (organisation or team characteristics, patient population, co-interventions and implementation process).

All focus groups and interviews were independently coded by two researchers (RBn and TM) to identify pertinent concepts and ideas by mapping to the CP-FIT domains. The first transcript was independently read by RBn and TM to generate initial codes and themes to match the framework, which were then compared and refined. Differences were resolved by negotiation and consensus, and consultation with a third researcher (JMN). A further transcript was coded using this schema, compared and refined, and this process was repeated until all transcripts were coded. Data were managed using NVivo V.12 (QSR International, Doncaster, Australia).

## 5. Conclusions

This study demonstrated that implementing a QI program that employed (a) a CDS tool to facilitate appropriate antibiotic prescribing by providing easy access to guidelines, and prompting for an indication in general practice, (b) auditing with automated data extraction and evaluation of appropriateness and (c) feedback, is not an easy undertaking. Barriers such as cost and practice access, program fitting into the GP workflow, data accuracy, and perception of the value of the program can hinder the success of the QI program. Many technical issues could be resolved with a more centralised approach to deployment beyond the pilot. However, an enthusiastic practice-wide approach, positive practice champions, and the passion to reduce inappropriate antibiotic prescribing can contribute to the success of the program. While the pandemic was an unexpected hindrance to this study, by ensuring all participating GPs understand the expectations of the program, allowing extra cost and time for the implementation of the tools, and identifying and involving practice champions in the development of the QI programs could address some of these barriers found and enable the successful implementation of future QI programs in general practice.

## Figures and Tables

**Figure 1 antibiotics-10-00867-f001:**
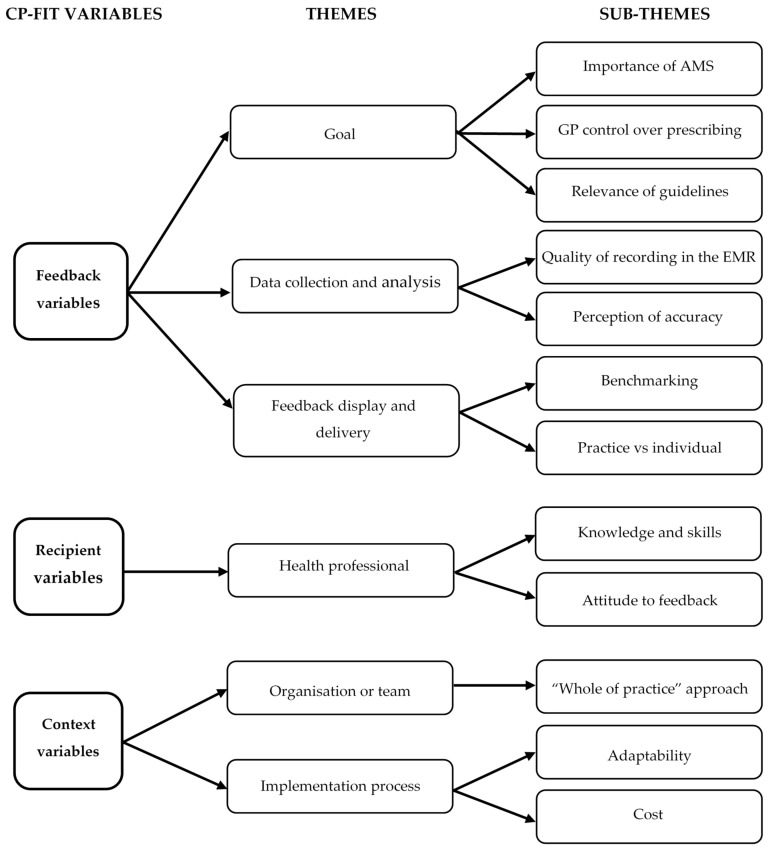
Clinical performance feedback intervention theory (CP-FIT) variables with corresponding themes and sub-themes. AMS = antimicrobial stewardship; EMR = electronic medical record.

**Table 1 antibiotics-10-00867-t001:** Demographics of GPs and practice managers (PM) participated in the focus groups and interviews.

Participant ID	Age (Range in Years)	Gender	Years of Experience in General Practice
GP1	31–40	Female	5
GP2	41–50	Female	18
GP3	41–50	Male	20
GP4	41–50	Female	16
GP5	51–60	Female	33
GP6	51–60	Female	23
GP7	51–60	Male	30
GP8	41–50	Male	15
GP9	41–50	Male	18
GP10	31–40	Male	4
GP11	31–40	Female	8
PM1	51–60	Female	17
PM2	51–60	Female	12
PM3	41–50	Female	5

**Table 2 antibiotics-10-00867-t002:** Number of antibiotics prescribed, number of indications documented in electronic medical record (EMR), number of indications entered in audit pop up tool, top three common indications, and top three prescribed antibiotics during pre- and post-audit periods from three GP practices.

Key Comparison Measures	Pre-Intervention	Post-Intervention
Survey Period	14–27 November 2019	17–31 August 2020
Number of antibiotic prescriptions	231	73
Indications documented in EMR	59% (*n* = 137)	62% (*n* = 45)
Indication entered in audit pop up tool	41% (*n* = 94)	38% (*n* = 28)
Top three common indications	Skin and soft tissue infection (27%)	Skin and soft tissue infection (33%)
	Ear nose and throat 1 (23%)	Acute cystitis (22%)
	Respiratory infection 2 (13%)	Respiratory infection 2 (12%)
Top three prescribed antibiotics	Amoxicillin (15%)	Amoxicillin-clavulanic acid (16%)
	Cefalexin (12%)	Cefalexin (15%)
	Doxycycline (8%)	Trimethoprim (11%)

^1^ Ear nose and throat includes acute otitis externa, acute otitis media, acute rhinosinusitis, pharyngitis and tonsillitis. ^2^ Respiratory tract infection includes infective exacerbation of asthma, bronchiectasis, bronchitis, influenza, pneumonia and upper respiratory tract infection not otherwise specified.

## Data Availability

Data can be available upon request.

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
