# Peer review of "Evaluating the Implementation of a Pilot Quality Improvement Program to Support Appropriate Antimicrobial Prescribing in General Practice"

_antibiotics, 2021, doi:10.3390/antibiotics10070867_

Round 1
Reviewer 1 Report
Biezen and colleagues addressed an emerging and eminent problem that needs to be overcome as quick as possible - antimicrobial resistance. Specifically, in this work, the authors evaluated the implementation of a pilot study targeting the application of a quality improvement program to support an appropriate antimicrobial prescribing in general practice. Indeed, antimicrobials, particularly antibiotics are indiscriminately prescribed, in most cases without prior antibiogram being requested.
This manuscript is in general well-written, with good scientific soundness and interest to the Antibiotics readers. However, there are some aspects that need to be carefully addressed:
- abstract: based on the findings stated in the last sentence of the introduction, what matters will be overcome with these advances?
- results: 1) l. 69: mean/median years of GPs experience should be added, and not merely presented in table 1; 2) Figure 1 legend should be added, specifically the full name of AMS and EMR; 3) please avoid the use of abbreviations, like "didn't" > change to did not, please check the whole manuscript; 4) l. 143-148: antibiograms are not routinely requested in Australian hospitals?; 5) considering that most patients are like a puzzle, composed of multiple traits and clinical conditions, the GPs mostly focus on the most important aspect and neglect the others? 6) do the authors think that the low appreciation of this program by some GPs may be related with their resistance to gets evaluated regarding clinical conduct? 7) what can the authors conclude on barriers of communication between the different professionals and even with patients? 8) based on data obtained, how the authors think to put this program in practice? do the authors think this program will be generally accepted, given the feedback of several GPs? what other strategies can be exploited to overcome some of the limitations/barriers found? would educational programs not be useful in this case?
- discussion: 1) l. 261 "de-value their profession" > continue update is a need; how the authors expect to overcome this main constrain? 2) do you think that antibiotics prescription is under finance pressure? how can you conclude about this? 3) l. 276-283: why the period was not extended?
- materials and methods: was pre-test intervention not designed neither applied?
- please standardize the use of abbreviations throughout the whole manuscript
Author Response
The authors would like to thank reviewer 1’s comments and feedback, and the time taken to review this manuscript.
We have addressed your comments point by point. Please see the attachment for full response.

Reviewer 2 Report
This is an interesting study focusing on the implementation of an AMS programm on GPs in Australia. Overall, this is a well conducted study.
It would be useful to explain how was the sample size calculated and the rational behind this. Why only 35 GPs were interviewed?How many GPs are there in Melbourne? Given the fact that payment was given for participation of GP practices could that affect the outcome?
Also a limitation section would be advisable
Author Response
This is an interesting study focusing on the implementation of an AMS programm on GPs in Australia. Overall, this is a well conducted study.
Response: The authors would like to thank Reviewer 2 for the valuable comments and feedback. We have revised the paper to address your comments.
It would be useful to explain how was the sample size calculated and the rational behind this. Why only 35 GPs were interviewed? How many GPs are there in Melbourne? Given the fact that payment was given for participation of GP practices could that affect the outcome?
Response: Thank you for your feed back. Our pilot study was conducted in three general practices included a total of 31 GPs. Of those, the participation rates were 100% (5 out of 5 GPs), 75% (21 out of 28 GPs) and 63% (5 out of 8 GPs) across the three practices. 11 GPs participated in the three focus groups. As this was a pilot implementation study, we had deliberately engaged three different general practices of different sizes. Due to the diversity of general practices, we did not intend or expect, for the results to be representative across all general practices. The practice incentive payment provided in this study was provided to subsidise practices for IT support, practice coordination, time needed to install both audit and CDs tools and testing and to compensate for the time taken to participate in the research component of the study. The actual time needed to implement this program was in excess of the payment that we provided. As such we feel it would unlikely affect the outcome.
Also a limitation section would be advisable
Response: Thank you for the feedback. We have included a limitation section. “The small number of participating practices was a limitation to our study. Data collected were limited to three general practices in metropolitan. Melbourne, and so our findings may not be generalizable to more diverse settings, including practices in rural or remote areas. In addition, our study may include selection bias as participants were more likely to have an interest in antimicrobial stewardship and QI programs. . The study was largely conducted during the COVID-19 pandemic, which was characterised by reduced respiratory tract infections presentations to general practice, significant changes to delivery of care more broadly (with transition to telehealth) and impacted the delivery and likely participation in the quality improvement activity. Despite this, all components of the intervention were delivered. We did not have access to additional funding in order to carry out additional audits after cessation of lockdown in Melbourne.” Please refer to page 8, lines 304-314.
Reviewer 3 Report
Thank you for the opportunity to review this manuscript which is a qualitative analysis of a QI project looking at implementing better antimicrobial stewardship tool. The qualitative data are not my area of expertise, hence I'm going to concentrate on some points that the authors could improve to bring out their arguments.
Although they have concentrated on the GPs and practice managers participating in the study and in the interviews, it would be very important to have a bit more data on the patient flows and characteristics. Currently, as the authors correctly state it's very difficult to make any judgement on the efficacy and safety of the QI intervention. It would be imperative to hev more data on the patients seen in the two periods across the three practices.
The following should be provided: number of face-to-face and remote consultations, the % breakdown of respiratory, other infectious and non-infectious consultations, the classes of ABx prescribed and some basic demographic characteristics of the patients. This could give the readers a clearer understanding, whether the QI project worked or if it is still indeed in it's infancy due to the pandemic disruption. When this data is provided in the results, the authors should put this into context with similar studies of similar healthcare systems.
Author Response
Thank you for the opportunity to review this manuscript which is a qualitative analysis of a QI project looking at implementing better antimicrobial stewardship tool. The qualitative data are not my area of expertise, hence I'm going to concentrate on some points that the authors could improve to bring out their arguments.
Response: The authors would like to thank Reviewer 3 for the valuable comments and feedback. We have revised the paper to address your comments.
Although they have concentrated on the GPs and practice managers participating in the study and in the interviews, it would be very important to have a bit more data on the patient flows and characteristics. Currently, as the authors correctly state it's very difficult to make any judgement on the efficacy and safety of the QI intervention. It would be imperative to have more data on the patients seen in the two periods across the three practices.
Response: In addition, we have included the following in the discussion: “While the GP NAPS audit data showed an overall decrease in the number of antibiotic prescribed, a slight overall percentage increase in the number of indications documented in the EMR, and an overall percentage decrease in the indication entered in the audit pop up tool, we cannot determine the significance of these results due to the small sample size.” Please refer to page 8, lines 270 to 273.
The following should be provided: number of face-to-face and remote consultations, the % breakdown of respiratory, other infectious and non-infectious consultations, the classes of ABx prescribed and some basic demographic characteristics of the patients. This could give the readers a clearer understanding, whether the QI project worked or if it is still indeed in it's infancy due to the pandemic disruption. When this data is provided in the results, the authors should put this into context with similar studies of similar healthcare systems.
Response: Thank you for your feedback. We have included the top three common infections and the top three antibiotics prescribed pre- and post- intervention in Table 2 – page 3, line 90. However, as this is a pilot implementation study to evaluate the quality improvement program in general practice, we did not focus on individual patient data but rather aggregate patient data across the three practices.
Round 2
Reviewer 1 Report
All comments raised were properly addressed. The only aspect I would like the authors change is the last sentence of introduction (l. 25-28). "General practice" is repeated
Reviewer 3 Report
Thank you for addressing my points.